# The effectiveness of layer-by-layer training using the information bottleneck principle

## Abstract

The recently proposed information bottleneck (IB) theory of deep nets suggests that during training, each layer attempts to maximize its mutual information (MI) with the target labels (so as to allow good prediction accuracy), while minimizing its MI with the input (leading to effective compression and thus good generalization). To date, evidence of this phenomenon has been indirect and aroused controversy due to theoretical and practical complications. In particular, it has been pointed out that the MI with the input is theoretically infinite in many cases of interest, and that the MI with the target is fundamentally difficult to estimate in high dimensions. As a consequence, the validity of this theory has been questioned. In this paper, we overcome these obstacles by two means. First, as previously suggested, we replace the MI with the input by a noise-regularized version, which ensures it is finite. As we show, this modified penalty in fact acts as a form of weight-decay regularization. Second, to obtain accurate (noise regularized) MI estimates between an intermediate representation and the input, we incorporate the strong prior-knowledge we have about their relation, into the recently proposed MI estimator of Belghazi et al. (2018). With this scheme, we are able to stably train each layer independently to *explicitly* optimize the IB functional. Surprisingly, this leads to enhanced prediction accuracy, thus directly validating the IB theory of deep nets for the first time.

## 1 Introduction

Deep neural nets (DNN) have shown unparalleled success in many fields, yet the theoretical understanding of these models is lagging behind. Among the various attempts to shed light on this field, Tishby & Zaslavsky (2015) have recently suggested a link between DNN training and the Information Bottleneck (IB) principle of Tishby et al. (1999). Their theory claims that optimal performance is attained if *each layer* simultaneously attempts to maximize its mutual information (MI) with the target space, while minimizing its MI with the input space. The logic is that the DNN layers should compress the latent representation so as to allow for good generalization, while retaining only the information essential for target prediction. Moreover, it was hypothesized that training DNNs through minimization of the popular cross-entropy loss, implicitly achieves this goal. More formally, it was suggested that during training, each DNN layer approaches the *maximal* value of the IB Langragian

$$\mathcal{L}_{\text{IB}} = I(Y; L_i) - \beta I(X; L_i), \tag{1}$$

where $I$ denotes MI, $X$ and $Y$ are the input and target random variables, $L_i$ is the latent representation at the output of the $i$th hidden layer, and $\beta$ is a trade-off parameter.

This elegant connection between DNN training and the IB principle was later supported by empirical experiments conducted by Shwartz-Ziv & Tishby (2017), as well as by others (Alemi et al., 2018; Belghazi et al., 2018; Kolchinsky et al., 2017; Saxe et al., 2018). These works either (i) showed the effect of optimizing the IB functional in (1) for classification tasks, or (ii) analyzed the dynamics of each layer in the information plane (defined by axes $I(X; L), I(Y; L)$) during training with the cross-entropy loss. And indeed, these works have shown such a scheme is beneficial in terms of classification accuracy, and that the IB functional tends to increase during training with the cross-entropy loss. Yet in all these demonstrations, nets were trained in an end-to-end fashion, where the IB objective was either enforced only on *a single layer* $L_i$, or not at all (trained only with a cross-entropy loss). Therefore, it remains unclear whether directly optimizing the IB functional of each

layer, as suggested by Tishby & Zaslavsky (2015), would indeed yield useful representations. The goal of this paper is to experimentally examine this hypothesis.

In addition to the lack of explicit validation, criticism on the rationale behind the IB theory of deep learning has surfaced lately. One major concern highlighted in the recent works of Saxe et al. (2018) and Amjad & Geiger (2018), is that for *deterministic* DNNs with continuous inputs, the term $I(X; L_i)$ is always infinite so that the IB principle is in fact meaningless. The authors suggest to resolve this by resorting to *stochastic* DNNs via the introduction of noise after each layer, which ensures that $I(X; L_i)$ is finite. And indeed, all attempts to train DNNs with the IB objective thus far employed stochastic nets (Alemi et al., 2018; Belghazi et al., 2018; Kolchinsky et al., 2017). Yet, deterministic DNNs are far more popular for most tasks, raising questions on the applicability of the IB theory for these models.

In this paper, we focus on validating (a regularized version of) the IB theory to deterministic DNNs trained on high-dimensional data. To alleviate the explosion of MI with the input space, we introduce noise *only for quantifying* the MI between the input $X$ and hidden layer $L_i$ (as also suggested by Saxe et al. (2018)), and not into the DNN itself. Namely, we replace the first term in (1) by $I(X; L_i + \varepsilon)$, where $\varepsilon$ is noise. As we show, the resulting term can be interpreted as a weight decay penalty, which aligns with common practice in DNN training. To estimate MI, we use an auxiliary net, similarly to the suggestion by Belghazi et al. (2018), and in contrast to applying a variational lower-bound as in (Alemi et al., 2018; Kolchinsky et al., 2017). However, as opposed to the original mutual information neural estimator (MINE) of Belghazi et al. (2018), which fails to accurately estimate $I(X; L_i + \varepsilon)$ in our setting[1], we tailor the architecture of the auxiliary net specifically for our case in which $L_i$ is a known deterministic function of $X$. This modification results in significantly more accurate estimates, as we show in Section 4.

Equipped with this estimation strategy, we directly test the validity of the IB theory of deep learning for deterministic models. This is accomplished by training a DNN while guaranteeing that *each* layer optimizes the IB functional in (1). Starting from the first hidden layer, each layer is trained independently, and frozen before moving on to the next layer. We find that such a training process is not only useful, but surprisingly consistently leads to slightly better performance than end-to-end training with the cross-entropy loss. This is thus the first explicit confirmation of the theory suggested by Tishby & Zaslavsky (2015), subject to adapting the definition of $I(X; L)$ so as to make this quantity meaningful. Please note that whether layer-wise training with the IB principle is generally beneficial over other layer-wise training schemes, or over other regularized end-to-end schemes, is out of this paper's scope.

Accompanying the original theory were several key observations, the most notable being the emergence of two-phase training dynamics (Shwartz-Ziv & Tishby, 2017). It was hypothesized that the second "compression" phase is responsible for enhancing the generalization capability. Yet Saxe et al. (2018) contradict this claim, showing that this phase may not exist when training with a cross-entropy loss. Our experiments align with those of Saxe et al. (2018), though we do observe these two phases when training explicitly with the IB functional. Surprisingly, when training with cross-entropy *and* weight decay regularization, the two-phase dynamics are also apparent. This suggests that this two-phase pattern is induced by regularization, and is thus indeed linked to generalization.

## 2    RELATED WORK

**Deep neural nets and The Information Bottleneck**    The Information Bottleneck (IB) technique for summarizing a random variable $X$ while maintaining maximal mutual information with a desirable output $Y$, was first introduced by Tishby et al. (1999). It is designed to find the optimal trade-off between prediction accuracy and representation complexity, by compressing $X$ while retaining the essential information for predicting $Y$. The IB optimization problem can be directly solved in discrete settings, as well as in certain simple families of continuous distributions, like jointly Gaussian random variables (Chechik et al., 2005). However, in general high-dimensional continuous scenarios, exact optimization becomes impractical due to the intractable continuous MI.

The IB principle was recently linked to DNNs by Tishby & Zaslavsky (2015), which hypothesized that DNN layers converge to the optimal IB curve during training. The subsequent work by Shwartz-

---

[1] For example, it does not tend to infinity when the noise is taken to zero.

Ziv & Tishby (2017) attempted to experimentally verify this by analyzing DNNs trained with the cross-entropy loss (with no regularization). In this work, the MI estimation was made tractable by binning (discretization) of the latent representations $L_i$, which works for "toy" examples but does not scale to real-world scenarios. Among several claims, the authors report that two training phases emerge: a "fitting" phase followed by a "compression" phase, which to their understanding is linked to increased generalization. Yet Saxe et al. (2018) contradict these claims, by linking the compression phase to the activation type and discretization strategy, and questioning the connection between this compression phase and generalization. Moreover, these authors as well as Amjad & Geiger (2018), also recognize that the term $I(X; L)$ in the IB functional is theoretically infinite for deterministic DNNs with a continuous input $X$, and thus the attempt to measure it is meaningless. Both works propose to remedy this by adding noise, which ensures this term is finite.

Despite these obstacles, DNNs were in fact trained on real high-dimensional data for classification tasks with the IB functional. The difficulties arising from the intractable and infinite term $I(X; L)$ were overcome by (i) training *stochastic* DNNs which ensure it is finite, and (ii) using a variational approximation of the MI which makes it tractable (Alemi et al., 2018; Kolchinsky et al., 2017; Chalk et al., 2016; Achille & Soatto, 2018; Belghazi et al., 2018). These schemes all rely on some form of injected stochasticity, and in fact, most enforce the IB objective only on a *single* "bottleneck layer". An attempt to optimize the IB functional for deterministic DNNs in a layer-wise fashion, as in the original theory, has yet to appear due to these practical difficulties. Here, we report on such a training scheme and present its results in the following sections.

We note that layer-wise training with a related information-theoretic objective has been studied by Ver Steeg & Galstyan (2015) and Gao et al. (2018), which have shown an effective and practical method to compose and analyze hierarchical representations. These works consider an *unsupervised* setting, whereas we analyze supervised DNN training (specifically) with the IB functional.

**Estimating mutual information**  Quantifying the MI between distributions is inherently difficult (Paninski, 2003), and is tractable only in discrete settings or for a limited family of problems. In other more general settings the exact computation is impossible, and known approximations do not scale well with dimension and sample size (Gao et al., 2015). Most recently, Belghazi et al. (2018) proposed the MI Neural Estimator (MINE) for approximating the MI between continuous high-dimensional random variables via back-prop over a DNN. The core idea is to estimate the Kullback-Leibler (KL) divergence that is used to define MI, through the maximization of the dual representation of Donsker & Varadhan (1983).

Minimizing the MI between the input $X$ and hidden layer $L_i$ of a DNN using this neural MI estimator can be efficiently accomplished, by formulating a minmax objective between the examined DNN and the estimator (an auxiliary net), similar to adversarial training (Goodfellow et al., 2014) (see Belghazi et al. (2018)). We use this strategy to enforce the IB objective on DNN layers during training. Note that the original authors also demonstrate the IB principle with this estimator, however they only do so on a single "bottleneck layer", by using the cross-entropy loss as an approximation for the MI with the desired output $Y$, and in an end-to-end manner (as described above).

## 3 THE NOISE-REGULARIZED MUTUAL INFORMATION PENALTY AND ITS RELATION TO WEIGHT DECAY

The intuitive goal of the penalty $I(X; L_i)$ in the IB principle, is to induce a "compressed" latent representation which does not contain information irrelevant for predicting the labels $Y$. However, for deterministic DNNs, if $X$ is continuous, then the MI between $X$ and any DNN layer $L_i$ is infinite and therefore meaningless. We seek a related measure of representation compactness which is more effective. One such alternative is to penalize for the MI between the layers *up to* an additive Gaussian noise $\varepsilon$, i.e. $I(X; L_i + \varepsilon)$. Note that noise is added *only* for the sake of quantifying complexity, and is not part of the net. Regularization by injection of noise is common practice in neural net training (Srivastava et al., 2014; Kingma et al., 2015; Wan et al., 2013; Poole et al., 2014), and was also analytically studied for the Gaussian IB case (Chechik et al., 2005).

To understand the effect of replacing the penalty $I(X; L_i)$ by $I(X; L_i + \varepsilon)$, we can write

$$I(X; L_i(X) + \varepsilon) = h(L_i(X) + \varepsilon) - h(L_i(X) + \varepsilon | X), \tag{2}$$

where $h$ denotes differential entropy, and we wrote $L_i(X)$ to emphasize that the latent representation is a deterministic function of $X$. First, notice that at zero noise level the second term becomes minus infinity (since $p_{L_i(X)|X}$ becomes a delta function), which highlights again the inadequacy of the penalty $I(X; L_i)$ for deterministic DNNs. Second, notice that the second term in (2) can be further simplified (for Gaussian noise) as

$$h(L_i(X) + \varepsilon | X) = \tfrac{1}{2} \log \left( \det(2\pi e \Sigma) \right), \tag{3}$$

where $\Sigma$ is the noise covariance matrix. This term is *independent of the DNN parameters*. This shows that the MI penalty $I(X; L_i(X) + \varepsilon)$ is in fact a penalty only on the entropy $h(L_i(X) + \varepsilon)$ of the representation $L_i$ (up to additive noise), which is a typical measure of compactness.

To develop further intuition, it is instructive to examine the simple case in which both the input $X$ and the additive noise $\varepsilon$ are scalar (independent) Gaussian variables and the transformation is linear, i.e. $L_i(X) = aX + b$. In this setting, simple calculation shows that

$$I(X; L_i + \varepsilon) = \frac{1}{2} \log \left( 1 + \frac{\sigma_X^2}{\sigma_\varepsilon^2} a^2 \right), \tag{4}$$

where $\sigma_X^2$ and $\sigma_\varepsilon^2$ are the variances of the input and noise, respectively. This function is monotonically increasing in $|a|$ and attains its minimum at $a = 0$, in which case $X$ and $L_i$ become independent. This highlights the fact that this term, in essence, induces a type of weight decay regularization, as popularly employed in deep net training. Clearly, the same intuition is valid also for non-linear high-dimensional settings, since weight decay (at the extreme) drives $X$ and $L_i$ to become independent and thus to have zero MI. Yet the precise form of the penalty is generally different between the MI regularizer and hand-crafted ones. Also notice that in our case, the noise level $\sigma_\varepsilon$ controls the strength of the regularizer, as we experimentally validate in Section 4.

In light of the above discussion, our goal is to explore the validity of the noise-regularized version of the IB Langragian,

$$\mathcal{L}_{\text{IB}} = I(Y; L_i) - \beta I(X; L_i + \varepsilon), \tag{5}$$

where $\varepsilon$ is white Gaussian noise.

## 4 ACCURATE ESTIMATION OF MUTUAL INFORMATION

Given a DNN, to estimate the two MI terms in (5), we rely on the recently proposed MI neural estimator (MINE) of Belghazi et al. (2018). This method uses the fact that $I(U; V) = \text{KL}(\mathbb{P}_{U,V} || \mathbb{P}_U \otimes \mathbb{P}_V)$ and exploits the dual representation of Donsker & Varadhan (1983),

$$I(U; V) = \sup_D \mathbb{E}_{(S,T) \sim \mathbb{P}_{U,V}} [D(S,T)] - \log \left( \mathbb{E}_{(S,T) \sim \mathbb{P}_U \otimes \mathbb{P}_V} [e^{D(S,T)}] \right), \tag{6}$$

where the supremum is over all functions $D : \mathcal{U} \times \mathcal{V} \to \mathbb{R}$ for which the second expectation is finite. To approximate this value, the expectations are replaced by sample means over a training set, and the optimization is performed over a smaller family of functions - those implementable by a deep net of predefined architecture. That is, $D$ is taken to be a DNN by itself, whose parameters are optimized via back-propagation with stochastic gradient descent. Note that to optimize Eq. (6), $D$ must learn to discriminate between (i) pairs $(u, v)$ drawn from the joint distribution $\mathbb{P}_{U,V}$, and (ii) pairs $(u, v)$ drawn *independently* from the marginal distributions $\mathbb{P}_U, \mathbb{P}_V$. This is similar to the principle underlying adversarial training (Goodfellow et al., 2014), where to estimate the Jenson-Shannon divergence between "real" and "fake" distributions, a critic DNN must learn to discriminate between "real" and "fake" data instances.

In our setting, we estimate the term $I(Y; L_i)$ with a MINE, as described above. However, estimating the term $I(X; L_i + \varepsilon)$ is typically far more demanding, due to its higher dimensionality and the continuous nature of both arguments. Experimentally, we observed inconsistent and non-convergent behaviors for this estimate. In particular, as illustrated in Fig. 2, the estimate does not grow indefinitely as the noise variance is taken to 0, despite the fact that the true MI becomes infinite in this case (see Sec. 3). To improve the estimation accuracy, we exploit the strong prior knowledge we have in our setting, which is that the latent representation $L_i$ is a known deterministic function of the input $X$. Specifically, $L_i = F_i(X)$, where $F_i$ is the function implemented by the first $i$ layers of the

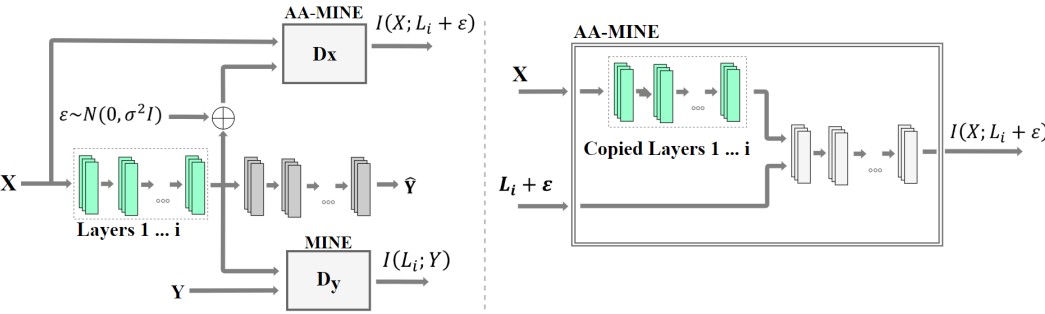

Figure 1: To measure the MI between the features at layer $i$ and the target labels $Y$, we use the MINE estimator of (Belghazi et al., 2018). To measure the MI between a noisy version of the features at layer $i$ and the input $X$, we use our AA-MINE, which contains an internal copy of layers $1, \ldots, i$ (see zoom-in on the right).

net. This implies that $D$ is actually trying to discriminate between (i) pairs $(x_1, F_i(x_1) + \varepsilon)$ where $x_1$ is drawn from $\mathbb{P}_X$, and (ii) pairs $(x_1, F_i(x_2) + \varepsilon)$ where $x_1, x_2$ are *independently* drawn from $\mathbb{P}_X$. Intuitively, the easiest way to achieve this is by applying the function $F_i$ on the first argument, and checking whether the result is close to the second argument, up to noise. However, unless the discriminator has the capacity to implement the function $F_i$ internally, it will fail to converge to such a rule. And even if its capacity suffices, convergence may be very difficult. Nevertheless, we can make the discriminator's task far easier, simply by informing it of the function $F_i$. We do this by implementing a copy of the sub-net $F_i$ within the discriminator and allowing it to pass its first input through this net (see Fig. 1). Since now the task becomes much simpler, we follow this by only a few fully-connected layers (see Appendix C), which suffice for obtaining accurate MI estimates and fast convergence, as can be seen in Fig. 2. Appendix A provides a detailed description of our AA-MINE training. Notice that for the limit case of zero noise level, the discriminator need only check whether the two inputs are identical, which can be perfectly accomplished with a very simple architecture. This drives the optimization of the objective in (6) to infinity, which aligns with the theory. We coin this MI estimator an *architecture aware* MINE (AA-MINE).

Before moving on to our primary goal, which is to *train* DNNs according to the IB objective, we can use our approach for the simpler task of just *measuring* the MI of each layer with the input/output while training with the common cross-entropy loss. To illustrate this, we trained a three-layer MLP with dimensions $784 - 512 - 512 - 10$ and ReLU activations on the (unmodified) MNIST dataset (LeCun et al., 1998) using the cross entropy loss, and measured the two MI terms in (5) for each layer during training (see Appendix C for details). As can be seen in Fig. 3(a), both terms tend to increase throughout the optimization process, and there is no apparent compression phase, in which the MI with the input suddenly begins to decrease. This phenomenon has also been observed by Saxe et al. (2018), and is in contradiction with the observations in (Shwartz-Ziv & Tishby, 2017), which seem to be associated with the binning they used to measure MI (Saxe et al., 2018). However, interestingly, when we *add weight decay* as a regularizer, the two-phase dynamics emerge (see Fig. 3(b)). As discussed in Sec. 3, weight decay induces a penalty on the MI with the input, and our experiments suggest that this penalty induces a compression phase which begins only after the MI with the output has reached high values. As weight decay is known to increase generalization (Krogh & Hertz, 1992), we conclude that the compression phase is indeed linked to generalization.

## 5 LAYER-BY-LAYER TRAINING WITH THE IB LOSS

The stage is now set for training a DNN layer-by-layer with the IB functional. Starting with the first hidden layer, our goal is to train each layer independently until convergence, freeze its weights, and then move on th the next layer. As our MI estimators are based on DNNs, we can achieve this goal by training them simultaneously with the layer of interest, by using back-propagation. Specifically, as in adversarial training and in the original MINE work (Goodfellow et al., 2014; Belghazi et al., 2018), to update the $i$th layer, we alternate between updating the discriminators $D_x, D_y$ according

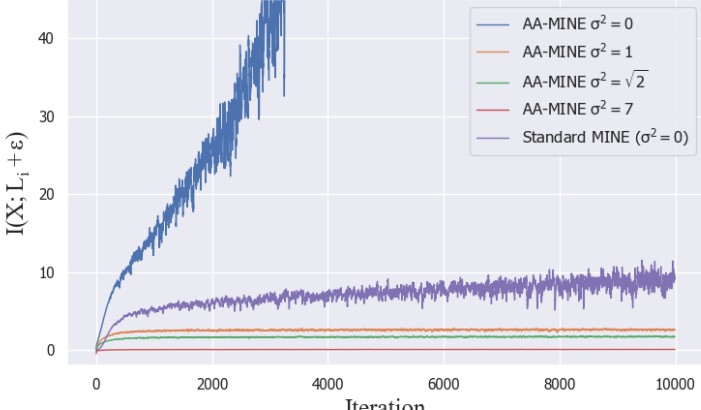

Figure 2: Estimating the (noise-regularized) MI between the input and last layer of a MLP ($784 - 512 - 512 - 10$) with random weights using a standard MINE (Belghazi et al., 2018) and our AA-MINE (see Sec. 4). For zero noise level ($\sigma^2 = 0$), the MI is theoretically infinite, which is captured by our AA-MINE but not by the standard MINE. This demonstrates the advantage of incorporating prior-knowledge into the MI estimation scheme. When enforcing stronger regularization (increasing noise level), our AA-MINE is stable and estimates decreasing values of MI, as expected.

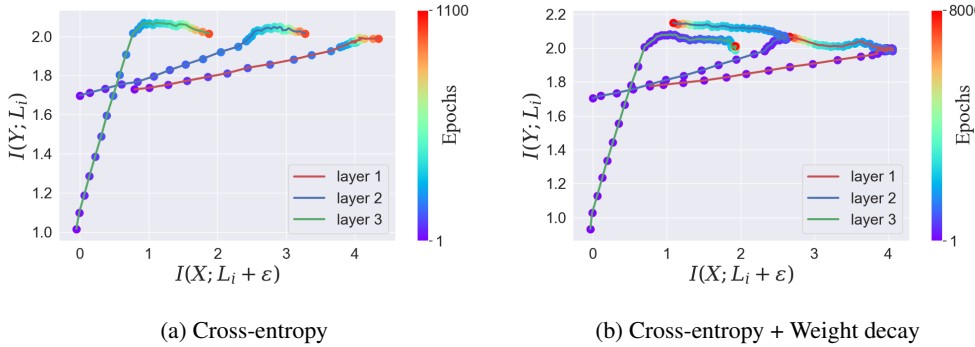

(a) Cross-entropy                   (b) Cross-entropy + Weight decay

Figure 3: Information plane dynamics with conventional training. In this experiment, a 3-layer MLP with ReLU activations was trained to classify MNIST digits and MI with input/target were measured with AA/standard MINE. (a) When using only the cross-entropy loss, both the MI with the input and the MI with the output tend to increase throughout the entire training process, aligning with the observations of Saxe et al. (2018). (b) When adding weight regularization, a compression phase emerges for the first two layers, where their MI with the input begins to decrease after their MI with the target reaches high values. Note that, as observed by Saxe et al. (2018), the noise-regularized complexity term does not satisfy the data-processing inequality, so that $I(X, L_i + \varepsilon)$ need not necessarily be larger than $I(X, L_j + \varepsilon)$ for $j > i$.

to (6) and updating the layer's parameters $\theta_i$ according to the noise-regularized IB objective (5) with the information terms replaced by the dual representation with the discriminator nets, namely

$$\max_{\theta_i} \ \left( \mathbb{E}[D_y(Y, L_i)] - \log(\mathbb{E}[e^{D_y(Y, L_i)}]) \right) - \beta \left( \mathbb{E}[D_x(X, L_i + \varepsilon)] - \log(\mathbb{E}[e^{D_x(X, L_i + \varepsilon)}]) \right). \quad (7)$$

Note that mutual information is invariant to (smooth) invertible transformations. Thus applying any invertible transformation on the net's output $\hat{Y}$, will not change its MI with $Y$. To simplify the classification task, it is convenient to seek an invertible transformation that brings $\hat{Y}$ closest to (the 1-hot representation of) $Y$. To achieve this, we *freeze the DNN parameters* after training is concluded, and post-train one additional fully-connected linear layer using a cross entropy loss.

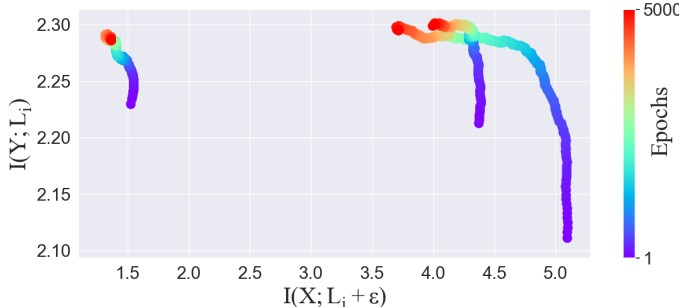

Figure 4: Information plane dynamics for layer-by-layer training explicitly with the IB functional. Two-phase training dynamics are apparent, and we observe a succeeding "compression" phase. Note that the data-processing inequality does not apply when estimating the *noise-regularized* MI (Saxe et al., 2018, App. C).

Table 1: Test-set accuracy on the (unmodified) MNIST dataset of a 3-layer MLP (784-512-512-10) trained layer-by-layer with the IB functional. Pushing each layer towards the optimal IB curve results in comparable performance to baselines, demonstrating the effectiveness of the IB principle. Notice that maximizing only the information with the output $I(Y; L_i)$ leads to inferior generalization, showing the importance of the regularization terms $I(X; L_i + \varepsilon)$.

| Train Set (%) | Training method | Test acc. (%) |
|:---:|:---|:---:|
| **1** | Cross-entropy loss (baseline) | 86.48 |
| | IB functional, only first term ($\mathcal{L} = I(Y; L_i)$) | 85.12 |
| | IB functional ($\mathcal{L} = I(Y; L_i) - \beta I(X; L_i + \varepsilon)$) | **86.57** |
| **100** | Cross-entropy loss (baseline) | 97.73 |
| | IB functional, only first term ($\mathcal{L} = I(Y; L_i)$) | 97.77 |
| | IB functional ($\mathcal{L} = I(Y; L_i) - \beta I(X; L_i + \varepsilon)$) | **98.09** |

This determines the best linear transformation to bring $\hat{Y}$ closest to $Y$, without changing the MI. Note that this does not expand the net capacity, since we use no non-linear activation between the last layer of the net and the post-trained fully-connected layer, so that they can be combined into a single linear layer after this post-training step.

Figure 4 depicts the information plane dynamics for IB-based layer-by-layer training of the same net of Fig. 3. Interestingly, in this experiment, we observe very different dynamics. First, the MI with the input does not increase at any stage during training. Second, here we have a clear two-phase process. Namely, each layer starts by moving upwards in the information plane to increase its MI with the desired output $Y$, and then turns to compress its latent representation by decreasing its MI with $X$, i.e. by moving leftwards. The resulting classification accuracies are reported in Table 1. Both when using 100% of the training set and when using only 1%, the performance of this training scheme is comparable to the baseline of training with the cross-entropy loss, and even slightly better. Also, notice that the penalty $I(X; L_i + \varepsilon)$ is essential for generalization, especially when using only 1% of the training set, and without out it, there is a drop in test accuracy. This is thus the first experiment to fully directly validate the effectiveness of the IB principle for DNNs.

We also test the IB principle with a conv-net for classifying the higher-dimensional CIFAR-10 dataset (Krizhevsky & Hinton, 2009). Here we train layer-by-layer with the IB functional a net with three conv-layers (16 filters, ReLU activations, max-pooling after each), which are followed by three fully-connected layers (512-512-10, ReLU activations; see Appendix C for full details). The resulting classification accuracies are reported in Table 2. In this more challenging scenario, the effectiveness of the IB principle is more pronounced. Training with the full IB functional leads to better accuracy compared to the cross-entropy baseline, and again, the regularization term which promotes a "compressed" latent representation proves quite advantageous.

Table 2: Test-set accuracy on the CIFAR-10 dataset for a conv-net (CONV16-CONV16-CONV16-FC512-FC512-FC10) trained layer-by-layer with the IB functional.

| Train Set (%) | Training method | Test acc. (%) |
|:---:|:---|:---:|
| **100** | Cross-entropy loss (baseline) | 58.23 |
| | IB functional, only first term ($\mathcal{L} = I(Y; L_i)$) | 60.59 |
| | IB functional ($\mathcal{L} = I(Y; L_i) - \beta I(X; L_i + \varepsilon)$) | **61.75** |

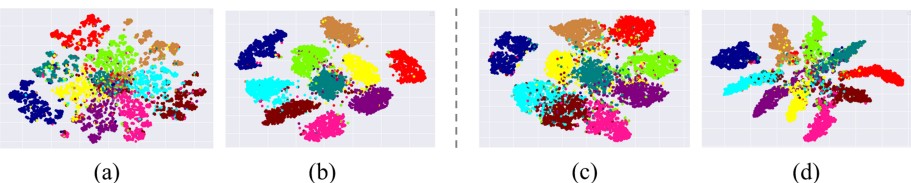

(a)      (b)      (c)      (d)

Figure 5: Visualizing the effect of training using the IB principle with t-SNE embeddings. *Left*: The embedding of $L_1$ when setting the trade-off coefficient to (a) $\beta = 10^2$, and (b) $\beta = 10^{-3}$. When increasing $\beta$, the training procedure attempts to discretize the latent representation, which indeed acts as a form of compression. *Right*: The embedding of $L_3$ when (c) training end-to-end with a cross-entropy loss, and (d) training layer-by-layer with the IB functional. With the IB functional, we obtain better separated and concentrated clusters.

It is interesting to try to decipher how layer-by-layer training with the IB loss affects the internal representations. In Fig. 5, we plot the t-SNE embeddings (Maaten & Hinton, 2008) of the latent representations $L_i$ of the MLP trained layer-by-layer on MNIST (Fig. 5(a),(b) with $100\%$ of the train set, Fig. 5(c),(d) with $1\%$). Recall that the regularization term $I(X; L_i + \varepsilon)$ enforces reduced entropy of the latent representation $L_i$ (see Section 3). The left side of Fig. 5 shows the effect of increasing/decreasing the influence of this term on the first hidden representation, $L_1$. As can be seen, for large $\beta$ values, the representation becomes quantized, which is a form of compression and indeed reduces the differential entropy by decreasing the effective support of the representation (differential entropy tends to $-\infty$ as the support of the distribution grows smaller). On the right side of Fig. 5, we show the difference between the representation $L_3$ at the output of the third layer when (c) training end-to-end with a cross-entropy loss and (d) training layer-by-layer with the IB principle. In the latter, we obtain better separated and concentrated clusters, which illustrates the advantage of directly enforcing the IB on each layer. To test this quantitatively, we compute the mean distance between clusters divided by the mean standard deviation of clusters, where higher values indicate better separation and concentration. The obtained value for (c) is 3.9, and for (d) is 5.2, confirming the advantageous outcome.

## 6 SUMMARY AND DISCUSSION

Our experiments demonstrate that training DNNs explicitly with the IB functional (and without a cross-entropy loss) leads to competitive and even enhanced prediction performance. This provides, for the first time, strong and direct empirical evidence for the validity of the IB theory of deep learning. Our training scheme was made possible by two key changes to prior attempts for training with the IB principle. First, as previously commented by Saxe et al. (2018), we used a noise-regularized version of the IB functional, which removes the theoretical difficulty of the MI between layers being infinite, while still being consistent with the intuition of a "complexity" penalty. As we showed, the resulting term acts as a form of weight-decay regularization. Second, we derived an MI estimation scheme specifically tailored for MI estimation between deterministic DNN layers. Our scheme is capable of *accurate* estimation in a scenario where this estimation has been shown to be generally difficult and usually intractable. We note that while the IB principle proved beneficial for layer-by-layer training, our alternating optimization procedure is computationally intensive. However, as we show in Appendix B, end-to-end training by enforcing the IB objective only on the last layer of the DNN, also leads to improved results, and is significantly more efficient.

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

# A    ALGORITHM FOR ESTIMATING NOISE-REGULARIZED MI

---

**Algorithm 1:** Estimating noise-regularized MI $I(X, L_i + \varepsilon)$

---

$b$ - batch size ; $\sigma$ - noise std ; $\mu$ - learning rate

**Input:** $X$ - input distribution ; $F_i$ - network up to layer i
**Output:** $I(X, L_i + \varepsilon)$

$\theta_0 \leftarrow$ Initialize the discriminator;
**for** $k = 1$ *to* $N$ **do**

1. Draw $b$ samples of X : $X^J = \{ x_J^{(1)}, x_J^{(2)}, ..., x_J^{(b)} \}$;
2. Draw $b$ samples of X : $X^M = \{ x_M^{(1)}, x_M^{(2)}, ..., x_M^{(b)} \}$;
3. Feed $X^J$ and $X^M$ through the network $(F_i)$, getting
   $L_i^J = \{ l_J^{(1)}, l_J^{(2)}, ..., l_J^{(b)} \}$ and $L_i^J = \{ l_M^{(1)}, l_M^{(2)}, ..., l_M^{(b)} \}$;
4. Generate noise samples $\varepsilon_J, \varepsilon_M \sim \mathcal{N}(0, \sigma^2 I)$ ;
5. Evaluate:
   $\mathrm{I}(\theta_k) = \frac{1}{b} \sum_{i=1}^{b} D_{\theta_{k-1}}(x_J^{(i)}, l_J^{(i)} + \varepsilon_J) - \log(\frac{1}{b} \sum_{i=1}^{b} e^{D_{\theta_{k-1}}(x_J^{(i)}, l_M^{(i)} + \varepsilon_M)})$;
6. $\theta_k \leftarrow \theta_{k-1} + \mu \cdot \nabla \mathrm{I}(\theta_k)$;

**end**

---

# B    TRAINING END-TO-END

Despite originally using the IB functional to train layer-by-layer, we can also use it to train the whole network end-to-end by taking into account only the MI of the classification layer with the target and input space. Adding the IB functional as a regularizer to the standard cross-entropy (CE) loss, we can perhaps prevent over-fitting in large scale DNNs. We therefore propose to minimize the objective

$$\mathcal{L} = \alpha\, \mathrm{CE}(Y, \hat{Y}) - \beta\, I(Y, \hat{Y}) + \gamma\, I(X, \hat{Y} + \varepsilon), \tag{8}$$

where CE is the standard cross-entropy loss.

Training DNNs end-to-end with the IB functional as regularizer has been proposed in several variations before (Belghazi et al., 2018; Alemi et al., 2018). Both of these works train stochastic encoder-decoder like networks to maximize cross entropy of decodings $\hat{Y}$ with the labels $Y$, which is a lower bound on MI $I(\hat{Y}, Y)$, while minimizing MI of stochastic encoding $Z$ with input space $X$. Alemi et al. (2018) estimate $I(Z, X)$ by using a Gaussian encoder and approximating $X$ by a Spherical Gaussian distribution of the same dimension as $Z$, while Belghazi et al. (2018) use a standard MINE and do not assume Gaussian distribution of neither $Z$ nor $X$. For deterministic DNNs, $I(Z, X)$ is infinite, making both techniques untranslatable to this more popular setting. Moreover, neither of these works is capable of dealing with high dimensional data for classification without pre-trained models, and so far have only shown results on MNIST.

Diverging from these works, we used our AA-MINE to estimate the MI $I(X, \hat{Y} + \varepsilon)$ and a standard MINE to estimate the $I(Y, \hat{Y})$, in addition to the standard cross-entropy loss. Since stochasticity is used only for measuring MI, and the AA-MINE needs only the classification layers representation of $X$, we can regularize deterministic DNNs trained on complex data and show the importance of explicitly measuring MI with target space, rather the only using a lower bound such as cross-entropy. Furthermore, this method can also be used with with stochastic methods to regularize DNNs, such as DropOut (Srivastava et al., 2014) or DropConnect (Wan et al., 2013). The results for MNIST and CIFAR-10 classification are reported in Table 3, and convergence curves are plotted in Fig. 6. See Appendix C for training details.

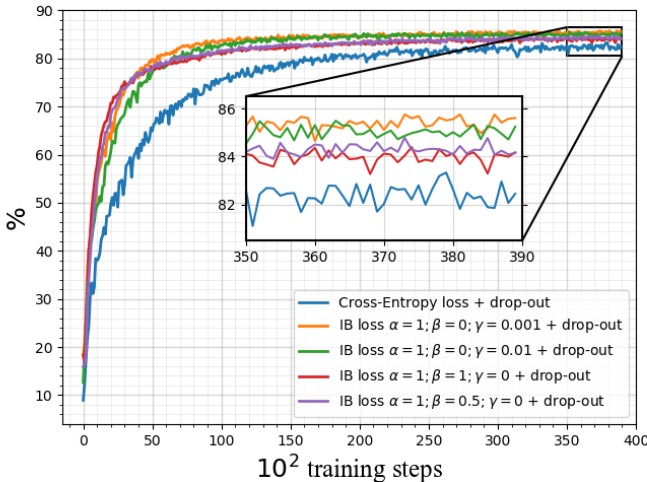

Figure 6: CIFAR-10 test accuracy as function of training steps for for various combinations of loss terms. Here we used a CNN with 6 convolutional layers and 3 fully connected layers (see Appendix C for details). In all cases, faster convergence to a superior result is obtained when using IB regularization relative to using only the cross-entropy loss. In particular, note that this demonstrates the superiority of explicitly enforcing high values of $I(\hat{Y}, Y)$ in conjunction with low cross-entropy values, over using cross-entropy alone.

Table 3: MNIST and CIFAR-10 end-to-end classification accuracy. Here we used 5000/20000 training steps for MNIST/CIFAR-10, respectively, and in each step we used 10 update steps for each of the two MINE discriminators. The variance of the regularization noise is $\sigma_\epsilon^2 = 2$.

| Data Set | Architecture | Train Set (%) | Loss (for minimization) | Test Acc. (%) |
|---|---|---|---|---|
| **MNIST** | MLP 784-600-400-10 | **100** | $\mathrm{CE}(Y,\hat{Y})$ | **97.73** |
| | | | $-I(Y;\hat{Y}) + 10^{-3} \cdot I(X;\hat{Y} + \varepsilon)$ | 97.87 |
| | | | $\mathrm{CE}(Y,\hat{Y}) - I(Y;\hat{Y}) + 10^{-3} \cdot I(X;\hat{Y} + \varepsilon)$ | **98.4** |
| | | **1** | $\mathrm{CE}(Y,\hat{Y})$ | **86.48** |
| | | | $-I(Y;\hat{Y}) + 10^{-3} \cdot I(X;\hat{Y} + \varepsilon)$ | 87.49 |
| | | | $\mathrm{CE}(Y,\hat{Y}) - I(Y;\hat{Y}) + 10^{-3} \cdot I(X;\hat{Y} + \varepsilon)$ | **88.11** |
| **CIFAR-10** | Conv-NN with DropOut | **100** | $\mathrm{CE}(Y,\hat{Y})$ | **83.90** |
| | | | $\mathrm{CE}(Y,\hat{Y}) - 0.5 \cdot I(Y;\hat{Y})$ | 84.6 |
| | | | $\mathrm{CE}(Y,\hat{Y}) + 10^{-3} \cdot I(X;\hat{Y} + \varepsilon)$ | **85.75** |

As can be seen, while the IB loss alone already leads to slight improvement, its combination with the CE loss provides a substantial boost. This is especially evident in cases where the training set is small w.r.t. the complexity of the problem.

## C  TRAINING DETAILS FOR THE MNIST AND CIFAR-10 EXPERIMENTS IN SECTIONS 4, 5, APPENDIX. B

In these experiments, we started by training both MINE discriminators $D_x, D_Y$ separately until convergence. Then, layer-by-layer training was performed with a total of 5000 iterations for each layer. In each of these iterations, we alternate between 1 step for updating the trained layer, and 10 steps for updating each of the MINE discriminators. For both MNIST and CIFAR-10, the variance

of the Gaussian noise $\varepsilon$ was $\sigma_\varepsilon^2 = 2$, and the learning rate was $10^{-4}$ for the MINE discriminators and $10^{-3}$ for the the DNN, respectively. For MNIST, the bottleneck parameter was $\beta = 10^{-3}$ for all the layers (Table 1). For CIFAR-10, the bottleneck parameter was $\beta = 0$ for the convolutional layers and $\beta = 10^{-3}$ for the fully-connected layers (Table 2).

For the end-to-end training experiment (Table 3), the MNIST architecture was a MLP ($784 - 600 - 400 - 10$) with ReLU activations, and the CIFAR-10 architecture was a CNN (CONV48-CONV48-CONV96-CONV96-CONV192-CONV192-FC512-FC512-FC10) with $2 \times 2$ max-pooling every second convolutional layer with $0.25/0.5$ drop-out probability on convolutional/fully-connected layers, respectively.

The architecture of the MINE discriminator was an MLP with two hidden layers of $1500$ neurons and Leaky-ReLU activations, and a final linear layer to a single neuron. The input was always taken to be a concatenation of the two variables between which MI is measured. For AA-MINE, we used the same architecture, only after passing the first input through a copy of layers $1 \dots i$ of the primary DNN. Therefore, this architecture was always applied to $F_i(X)$ and $F_i(X) + \epsilon$ in the case of AA-MINE.

In all experiments, training was performed with the Adam optimizer (Kingma & Ba, 2015) via python-TensorFlow (Abadi et al., 2015).

