# OpenReview forum: "The effectiveness of layer-by-layer training using the information bottleneck principle"
_ICLR.cc/2019/Conference_

### Official Review · AnonReviewer2 · 2018-11-03
**Layer-wise explicit IB functional training in DNNs**

**Rating:** 5
**Confidence:** 5

**Review:**

This paper provides a method to do explicit IB functional estimation for deep neural networks inspired from the recent mutual information estimation method (MINE).  By using the method, the authors 1) validate the IB theory of deep nets using weight decay, and 2) provides a layer-wise explicit IB functional training for DNN which is shown to have better prediction accuracy.

Pros:
- The paper carefully constructs a method to estimate the mutual information between high dimensional variables and address the infinite mutual information issue by adding noise to the output. This is novel and theoretical sounding.
- The paper connects the IB theory of DNN with weight decay, which is a novel founding.

Cons:
- The paper claims no literature has been doing IB functional on a layer-by-layer objective, however, see [1, 2] for the total correlation explanation work which is closely related to IB functional and they have also verified the effectiveness of layer-by-layer objective.
- The scope of the paper is unclear. It seems that the paper is trying to convince two things to the readers: 1) The compression phase in DNN does exist 2) Layer-wise training helps to improve the accuracy. Although these two things are close related to each other (because they all requires to estimate the IB functional), it seems that neither these two conclusions are convincing. First, the compression phase is achieved only through weight decay; without weight decay, as shown in the paper, the compression phase is gone. Does that verify the incorrectness of IB theory of deep nets? Second, for the layer-wise training, the paper only compares the layer-wise IB objective with the cross entropy loss. But if we really want to show the `effectiveness` of `layer-wise` training, one should compare the `layer-wise` training with `end-to-end` training while keeping the objective itself fixed. Otherwise, it is really difficult to draw conclusions about why the accuracy is improving, it is because of the objective changes or because of the `layer-wise` training.
- How does the beta (in IB objective) selected in the experiments for comparison? Do you use a validation dataset, and what is the final beta? If the paper fine-tune beta on the validation dateset, then the comparison of "IB functional, only the first term" and "IB functional" is unfair.

[1]  Ver Steeg et al. Maximally Informative Hierarchical Representations of High-Dimensional Data. AISTATS 2015
[2]  Gao et al. Auto-Encoding Total Correlation Explanation. Arxiv 1802.05822.

[update] After carefully reading the response (also from other reviewers), I decide not to change my rating.

---

> ### Author Response · Authors · 2018-11-09
> **Response**
>
> Thank you for the review. The points raised will help us clarify and improve our paper. Specifically:
>
> - Works [1], [2], which we weren’t aware of, are indeed related to layer-wise training with a closely-related unsupervised variant of the information bottleneck. We have now added a brief discussion on these works. Thank you for pointing this out. Please note that these works do not validate the IB theory, though, because their objective is unsupervised.
>
> - Regarding the scope of our paper, let us try to clarify. The goal of our paper is to validate the theory proposed by Tishby and Zaslavsky (2015). This theory suggests that each DNN layer should ideally optimize the IB functional to reach good performance. This theory was only partially and implicitly validated beforehand. For the purpose of testing this theory directly, we adopt a layer-wise training scheme which shows that the theory is indeed sound. We agree that it is interesting to independently study the effect of the two factors: the objective modification and the shift to a layer-wise training scheme. However we believe this is out of the scope of this paper and leave this for future work. Our paper does not attempt to conclude that layer-wise training is advantageous in this scenario, although this certainly might be the case. We have now stated the paper scope more clearly in the introduction.
>
> - During our attempts to validate the original theory of Tishby and Zaslavsky (2015), we observed an interesting behavior regarding the controversial argument of (Shwartz-Ziv & Tishby 2017) regarding the existence of a “compression phase” when training with the cross-entropy loss. Particularly, this phase didn’t appear for “vanilla” cross-entropy training, but did appear for cross-entropy training with weight decay regularization. We therefore briefly reported this, and tried to provide some insight in continuation to previous works discussing this. This observation does not contradict the original theory of Tishby and Zaslavsky (2015), which only claimed that the IB loss should ideally be optimized in each layer, but it does contribute to the discussion on the observations in (Shwartz-Ziv & Tishby 2017). In any case, this is not the main focus of our paper. Our main goal is to explicitly validate the theory of Tishby and Zaslavsky (2015), and this we do by layer-wise training with the IB loss.
>
> - For the case of “IB functional, only first term”, beta is zero (beta multiplies the second term). For the “IB functional” case, beta was chosen from {10^-4, 10^-3, 10^-2, 10^-1} on a validation set. The chosen beta for each experiment appears in Appendix C. We do not see why this is unfair.

---

### Official Review · AnonReviewer1 · 2018-11-05
**Confused discussion, lacking experiments, strong reject.**

**Rating:** 2
**Confidence:** 4

**Review:**

While overall the writing quality of the paper is high, the paper itself is a strong rejection.  I believe the analysis of the paper is at points flawed, and the experiments are minimal.

This work attempts to study the degree to which a layer by layer information bottleneck inspired objective can improve performance, as well as generally attempt to clarify some of the discussion surrounding Shwartz-Ziv & Tishby 2017.  Here, the authors study a deterministic neural network, for which the mutual information estimation is difficult (I(X,L)) and error prone.  To combat this they use the noise-regularized mutual information estimator (I(X; L+eps)).  To actually estimate the mutual information the authors use the MINE estimator of Belghazi (2018).  Here they suggest using the neural network itself as a structural element in the form of the discriminator to take advantage of the specific circumstances in this case.  Doing this ensured that their estimator diverged in the zero noise limit as expected.  From here they show some experimental results of the effect of their objective on an MNIST / CIFAR10 classification task.

This paper fits into what is an increasingly large discussion in the literature, surrounding Information Bottleneck.  The paper itself does a very good job of citing recent relevant work.  Technically however I take issue with the framing of previous work in the last paragraph of the "Deep neural nets" subsection of Section 2.  Technically Achille & Soatto explicitly formed a variational approximation to the posterior over the weights of the neural network and so was not a "single bottleneck layer" as stated in the paper.  More generally at the end of that paragraph it is implied that the single bottleneck layer scheme "deviates from the original theory".  This is a misleading characterization of the original information bottleneck (Tishby et al 1999) in which there was a single random variable, a representation of the data (Z) satisfying the Markov conditions Z <- X -> Y.   I believe the authors instead meant to say that the cited works deviate from the information bottleneck theory of learning suggested in (Shwartz-Ziv & Tishby 2017).  In general the paper does a poor job of distinguishing between the Shwartz-Ziv & Tishby paper and the rest, but this is a distinction that should be maintained.  The original information bottleneck may and has demonstrated utility regardless of whether the information bottleneck generally can help explain why ordinary deterministic feed forward networks trained with cross entropy and sgd generalize well.

This also raises one of the main problems with the current work. The title, abstract and especially the conclusion ("This provides, for the first time, strong and direct emperical evidence for the validity of the IB theory of deep learning") seem to present the paper as somehow offering some clarity and further support for the assertions of the Shwartz-Ziv & Tishby 2017 paper, but that paper hoped to establish that information bottleneck can explain the workings of ordinary networks.  Here the authors modify the ordinary cross entropy objective, and so their networks are necessarily not ordinary and so they cannot claim they have helped clarify our understanding of the vast majority of neural networks currently being trained.  Again, this is distinct and should be kept distinct from the utility of their proposed objective, itself inspired by the information bottleneck.  Here too the paper falls flat.  If instead of attempting to comment on networks as they are designed today they aim to proposed a new information bottleneck inspired objective they really ought to directly compare other attempts along those lines (such as the ones they themselves cite  Alemi et al. 2018, Kolchinsky et al. 2017, Chalk et al. 2016, Achile & Soatto 2018, Belghazi et al. 2018) but there are no comparative studies.

The experiments are extremely lacking, not only are any of their cited alternatives compared, they don't compare to what would be an equivalent network to their but where they did utilize the noise at every layer and actually made the network stochastic.  Their reported numbers are not very impressive with their top MNIST number at 98.09 and their baseline at 97.73. These numbers are worse than many of the papers they themselves cite.  Only a single comparative results for both a limited training set run and the full one are shown, as well as only a single choice of beta.  The CIFAR10 numbers are not very good either.  There is some discussion of the text suggesting they believe their method acts like an approximate weight decay, but there are no results showing the effect of weight decay just on the baseline classification accuracies they compare against.

Technically a deterministic function need not have infinite mutual information, if it is non-invertible, i.e. the sign function, or just floating point discretization.

Their own results in Figure 2 and the main body of the text highlight that the authors believe the true mutual information between the activations of the intermediate layers and the input is infinite.  If the true mutual information is infinite and the noise regularized estimator is only meant for comparative purposes, why then are the results of the training trajectories interpreted so literally as estimates of the true mutual information?

Just plugging in the Discriminator for the objective (equation (7)) is flawed.  The discriminator, if optimal would learn to approximate the density ratio 1 + log p(x,y)/(p(x) p(y)) .   ( see f-GAN, Norowin et al. 2016).  How does this justify using the individual elements of the discriminator in the functional form of the IB objective?

At the bottom of page 6 they rightfully say that mutual information is invariant to reparameterizations, but their noise regularized mutual information estimator is not (by their own reference (Saxe et al 2017).

The discussion at the center of page 8 is confusing.   They claim that Figure 5 (a) is more 'quantized' than (b) and "has reduced entropy".  I think it should be the other way.  More clusters should translate to a higher KL divergence, or higher entropy.  If you need only identify which cluster an activation is in, that should require log K nats where K is the number of clusters.  (a) shows more clusters and so seems like it should cost more and have a higher entropy not a lower one.

Despite a recurring focus of the text that this paper applies and information theoretic objective at each layer of the network, and hence is novel, the final sentence of the paper suggests it might not actually be needed and single layer IB objectives can work as well.

---

> ### Author Response · Authors · 2018-11-09
> **Response - part I**
>
> Thanks for the comments. We may have not clarified the scope and goals of our paper well enough, as well as the sense in which the effectiveness of layer-by-layer training via the IB objective helps explain the theory of (Tishby & Zaslavsky, 2015) (not that of their 2017 paper, which specifically focused on nets trained with the cross-entropy objective). Let us try to clarify these points below:
>
> - First, we do not attempt to suggest a novel IB-based objective for enhancing DNN accuracy. Also, we do not attempt to validate whether DNNs trained with the cross-entropy loss do or do not optimize the IB objective, as in (Shwartz-Ziv & Tishby 2017). Our sole goal is to examine the original IB theory of DNNs suggested by Tishby and Zaslavsky (2015), and to do so for deterministic DNNs (which are more popular than stochastic ones). The principal claim in this theory is that ideally each layer should optimize the IB objective. This has not yet been directly validated to date. What has been observed empirically in (Shwartz-Ziv & Tishby 2017) and in later works, is that when training nets with the cross-entropy loss, the IB objective tends to improve along the iterations. Combining this with the fact that training with the cross-entropy loss leads to good performance, one could claim that this indicates that a good IB loss in each layer leads to good performance. However, due to the difficulties in measuring MI (and the fact that I(X,L) is theoretically infinite in most interesting cases) these experiments do not give a good indication of whether the IB objective has indeed reached an optimal value  in each layer. Therefore, to directly examine the hypothesis of Tishby and Zaslavsky (2015) that optimizing the IB objective in each layer is the “ideal” thing to do in some sense, here we take “ordinary” DNNs and optimize each layer using the IB objective, with no cross-entropy term. Surprisingly, we find that this leads to competitive (and even enhanced) accuracy and generalization, and therefore constitutes a direct validation of the theory by Tishby and Zaslavsky (2015). We hope this clarifies the scope and logic behind our arguments. We have modified the introduction to make the scope and goal of our paper more clear.
>
> - Regarding comparisons, we don’t compare the accuracy of our scheme to other IB-based methods or regularization techniques, because our paper does not propose a novel training algorithm or regularization term. We do not claim that our IB-based layer-by-layer training scheme is practical for large-scale problems, or better than other schemes of similar flavor, and do not advocate its use in applications. As mentioned above, our experiments are only meant to explicitly validate the IB theory of deep learning for deterministic DNNs. The fact that this scheme leads to performance which is on par with end-to-end cross entropy training is the key point in the experiments. Note that the accuracy achieved in our experiments doesn’t compare to other works as they used DNNs with greater capacity, thus comparing these numbers is beside the point.
>
> - The paper by Achille & Soatto (2018) indeed does not fall into the category of “single bottleneck layer”, as you rightfully indicate. It is in the category of stochastic DNNs. This was a phrasing mistake. We have rephrased that sentence. In any case, as mentioned above, the scope of our paper is to study deterministic DNNs.
>
> - Regarding use of the term “original theory”, please note that by that we refer to the theory of Tishby and Zaslavsky (2015) which connects the IB principle to DNN training and not to the original IB principle (Tishby et al., 1999), which of course was not presented specifically in the context of DNNs. We have attempted to clarify this in the introduction, although we originally never referred to the 1999 paper in the context of DNNs within the text (it was cited only twice very briefly in the intro and related work sections).
>
> - If X is a continuous RV and f is a deterministic function, then it is a known fact that the mutual information between X and f(X) is always infinite. This is true regardless of whether f is invertible or not (even in the extreme case where f(X)=0). Please see Eq. (2) and the sentences below. Specifically, I(X,f(X)) = h(f(X)) - h(f(X)|X), where the conditional differential entropy (the second term) becomes minus infinity when f(X) is a deterministic function of X. This happens since the conditional density function becomes a delta function, which is true for invertible and non-invertible functions alike. This is as opposed to the discrete case, in which H(f(X)|X)=0 and there is no problem of this sort. Therefore, it is meaningless to study the original IB principle for deterministic DNNs with a continuous input, which is why we propose to study a noise-regularized version of the IB principle for DNNs.

---

> ### Author Response · Authors · 2018-11-09
> **Response - part II**
>
> - Eq. (7) is indeed a typo, thank you for pointing this out. We certainly don’t maximize the objective in (7) where the discriminator output replaces the mutual information, but the Donsker-Varadhan dual representation (6) substituted into the IB functional (5). We fixed this in our revised paper.
>
> - Our noise regularized mutual information is not meant only for comparative purposes. We claim that this quantity is in fact a more appropriate measure for “compactness” or “complexity” than the mutual information itself (which is infinite). As such, we propose to replace the horizontal axis in the original “information-plane” of Tishby’s papers by the noise-regularized mutual information when it comes to analyzing deterministic DNNs.
>
> - Regarding Fig. 5, we understand the confusion, thanks for pointing this out. Note that this is a continuous representation, therefore we are talking about differential entropy and not (discrete) entropy. Now, differential entropy depends not only on the number of clusters, but also on their sizes. Even a single cluster can have a very large entropy, if its size is large. For example, the entropy of a Gaussian is 0.5 ln(|2 \pi \Sigma|), which can be arbitrarily large if we take the determinant of \Sigma to be large. The point is that the entropy of a continuous distribution becomes smaller as its (effective) support becomes smaller, eventually tending to minus infinity for a distribution that is supported on a finite discrete set of points. This is exactly what’s seen in Fig. 5. In (b) the clusters are very large and the effective support of the distribution is large, whereas in (a) the clusters are very small and the effective support of the distribution is small. Therefore the differential entropy of (b) is larger than that of (a). We have clarified this point in the paper.

---

### Official Review · AnonReviewer3 · 2018-11-07
**The effectiveness of layer-by-layer training using the information bottleneck principle**

**Rating:** 5
**Confidence:** 4

**Review:**


This work is about layer-wise training of networks by way of optimizing the IB cost function, which basically measures the compression of the inputs under the constraint that some degree of information with respect to the targets must be preserved. Both terms of the IB cost function are formalized as mutual informations, but since in neural nets, the latent "compression" is a deterministic function of the inputs, a severe technical problems arises: the joint distribution between p-dimensional inputs X and the q-dimensional latent compression L is degenerate in that  its support lies in a space of dimension p (and not p+q as it would be in the non-degenerate case). As a consequence, no p.d.f. exists (with respect to the Lebesgue measure of R^{p+q}). Thus, defining mutual information is cumbersome. The paper attempts to overcome this problem by using a noisy version of the latent compression, i.e. L' = L + \epsilon, which can be seen as an "ad hoc" fix of this problem. Not too surprising, this additive noise works as a ridge-type (or weight-decay) regularizer, just as a Gaussian prior in regression.

On one hand, I find this paper interesting, because it aims at carefully studying the proposed link between DNN training and IB optimization, thereby showing that layer-wise IB training indeed seems to work very well in practice. Such results are certainly interesting, both from a theoretical and from a practical point of view. On the other hand, I honestly think that on the conceptual side, this work does not make that many really interesting contributions. The observation that additive noise works as a weight-decay regularizer is in my opinion almost trivial, and any claims about experimental results "validating(!) the IB theory" seem to contain some degree of over-selling. In summary, I think that this is a paper that certainly contains some interesting ideas, but on the other hand I am not fully convinced about the significance and relevance of the findings.

---

> ### Author Response · Authors · 2018-11-09
> **Response**
>
> Thank you for your review, we are happy you found our work interesting. The IB theory of deep learning by Tishby and Zaslavsky (2015) has attracted much interest, and increasing controversy is arising regarding some of the provided insights, such as the existence of a compression phase in training with the cross-entropy loss. However, the very basic claim of this theory, that each layer should ideally optimize the IB functional, was never verified. A direct way to validate the correctness of this claim, is to explicitly optimize the IB functional of each layer, as we did. We believe that this key element of the theory is very important to validate experimentally, perhaps even more than understanding the exact information dynamics during training with the cross-entropy loss.

---

### Author Response · Authors · 2018-11-09
**New revision**

A revision of our paper has been uploaded, after incorporating the reviewers comments and suggestions.

---

### Meta-Review · Area_Chair1 · 2018-12-13
**Clever application of MINE, but unclear how strongly the results validate information bottleneck theory**

**Confidence:** 4
**Recommendation:** Reject

**Metareview:**

This paper does two things. First, it proposes an approach to estimating the mutual information between the input, X, or target label, Y, and an internal representation in a deep neural network, L, using MINE (for I(Y;L)) or a variation on MINE (for I(X;L)) and noise regularization (estimating I(X;L+ε), where ε is isotropic Gaussian white noise) to avoid the problem that I(X;L) is infinite for deterministic networks and continuous X. Second, it attempts to validate the information bottleneck theory of deep learning (Tishby and Zaslavsky, 2015) by exploring an approach to training DNNs that optimizes the information bottleneck Lagrangian, I(Y;L) − βI(X;L+ε), layerwise instead of using cross-entropy and backpropagation. Experiments on MNIST and CIFAR-10 show improvements for the layerwise training over cross-entropy training. The penalty on I(X;L+ε) is described as being analogous to weight decay. The reviewers raised a number of concerns about the paper, the most serious of which is that the claim that the layerwise training results validate the information bottleneck theory of deep learning is too strong. In the AC's opinion, R1's critique that "[i]f the true mutual information is infinite and the noise regularized estimator is only meant for comparative purposes, why then are the results of the training trajectories interpreted so literally as estimates of the true mutual information?" is critical, and the authors' reply that "this quantity is in fact a more appropriate measure for “compactness” or “complexity” than the mutual information itself" undermines their claim that they are validating the information bottleneck theory of deep nets because the information bottleneck theory claims to be using mutual information. The AC also suggests that if the authors wish to continue this work and submit it to another venue, they (1) discuss the fact that MINE estimates only a lower bound that may be quite loose in practice and (2) say in their experimental section whether or not the variance of the regularizing noise was tuned as a hyperparameter, and if so, how results varied with different amounts of noise. Finally, the AC regrets that only one reviewer participated in the discussion (in a very minimal way), despite the reviewers' receiving several reminders that the discussion is a defining feature of the ICLR review process.